# Role of heat shock protein 60 in primed and naïve states of human pluripotent stem cells

**Hong Seo Choi, Hyun Min Lee, Min Kyu Kim, Chun Jeih Ryu** *

Department of Integrative Bioscience and Biotechnology, Institute of Anticancer Medicine Development, Sejong University, Seoul, Korea

* cjryu@sejong.ac.kr

**Data Availability Statement:** All relevant data are within the paper and its Supporting Information files.

## Abstract

Human pluripotent stem cells (hPSCs) exist in at least two distinct states in mammals: naïve pluripotency that represents several molecular characteristics in pre-implantation epiblast and primed pluripotency that corresponds to cells poised for differentiation in post-implantation epiblast. To identify and characterize the surface molecules that are necessary for the maintenance of naïve hPSCs, we generated a panel of murine monoclonal antibodies (MAbs) specific to the naïve state of hPSCs. Flow cytometry showed that N1-A4, one of the MAbs, bound to naïve hPSCs but not to primed hPSCs. Cell surface biotinylation and immunoprecipitation analysis identified that N1-A4 recognized heat shock protein 60 (HSP60) expressed on the surface of naïve hPSCs. Quantitative polymerase chain reaction (qPCR) analysis showed that HSP60 expression was rapidly downregulated during the embryoid body (EB) differentiation of primed hPSCs. HSP60 knockdown led to a decrease in the expression of pluripotency genes in primed hPSCs. HSP60 depletion also led to a decrease in the expression of pluripotency genes and representative naïve-state-specific genes in naïve hPSCs. Taken together, the results suggest that HSP60 is downregulated during differentiation of hPSCs and is required for the maintenance of pluripotency genes in both primed and naïve hPSCs, suggesting that HSP60 is a regulator of hPSC pluripotency and differentiation.

## Introduction

Human pluripotent stem cells (hPSCs), which include human embryonic stem cells (hESCs) and human induced pluripotent stem cells (hiPSCs), can self-renew indefinitely in culture and become almost any cell type in the human body [1, 2]. In rodents, pluripotent stem cells (PSCs) exist in two states: naive pluripotency that represents the ground state of pluripotency found in the preimplantation epiblast stem cells and primed pluripotency that represents more committed cells found in the postimplantation epiblast stem cells [3]. Naive PSCs can be expanded clonally, can be readily engineered by homologous recombination, have unlimited differentiation potential, exhibit low level of global DNA methylation, and are able to produce teratomas and chimeras, whereas primed PSCs cannot be expanded clonally, are difficult to modify genetically by homologous recombination, have limited differentiation potential,

**Funding:** This study was supported by the National Research Foundation of Korea (2016M3A9C6918220 and 2018M3A9H1023139) to CJR. The funders had no role in study design, data collection and analysis, decision to publish, or preparation of the manuscript.

**Competing interests:** The authors have declared that no competing interests exist.

exhibit high level of global DNA methylation, and are unable to produce chimeras [3, 4]. Typical hPSCs in culture have the same characteristics as mouse epiblast stem cells derived from the postimplantation epiblast, although they are originally derived from the preimplantation epiblast [1, 5, 6]. Thus, hPSCs exhibit properties that resemble primed pluripotency, indicating that the current conventional culture conditions for hPSCs are not able to maintain hPSCs with naïve pluripotency. Therefore, many researchers have been developing various approaches to obtain naïve hPSCs in culture.

To date, multiple approaches, based on earlier mouse studies, have been developed to derive and maintain naïve hPSCs [7–20]. Naïve hPSCs have been generated by converting primed hPSCs to an earlier state, by inducing naive pluripotency in somatic cells and by culturing pre-implantation embryos. Generally, all these approaches require leukemia inhibitory factor (LIF), chemical inhibitors, and/or expression of transgenes. However, naïve hPSCs from different studies have different characteristics, although they have some characteristics of human inner cell mass cells. Naïve hPSCs can be divided into three states, bona fide/ground, formative/intermediate, and primed state, and most of the naive hPSC induction methods mainly generate the formative/intermediate state while 5iLAF and t2iLGo methods generate the bona fide/ground state [13, 14, 18, 21]. Thus, current conditions employed to generate naïve hPSCs show great variation in efficiency and characteristics, suggesting that it needs to identify a consensus standard for generating naïve hPSCs [22, 23].

To define and classify naïve hPSCs with various states, a systematic study of surface markers specific for naïve hPSCs is needed. Various surface markers would help with the identification and purification of naïve hPSCs from various heterogeneous populations and would be also important for sub-population analysis during the naive hPSC induction process. To date, there are no known cell-surface markers that could exclusively define naïve hPSCs [24, 25]. SSEA-3, SSEA-4, CD24 and CD90, cell surface markers for primed hPSCs, have been suggested to monitor naïve hPSCs because they are downregulated in naïve hPSCs [25]. CD7, CD75, CD77 and CD130 are also suggested to be naïve hPSC-specific surface markers because they are upregulated in naïve hPSCs [26]. However, they are also positive for the trophectoderm cells in human embryos [26]. Sushi containing domain 2 (SUSD2) is also reported as a cell surface marker to define naïve hPSCs, but it is not required to establish naïve pluripotency [27, 28]. Therefore, there remains a need to find more standardized surface markers that can positively define naïve hPSCs within a heterogeneous population.

To study naïve hPSC-specific surface markers, we converted established primed hPSCs to naïve hPSCs by culturing primed H9 hPSCs [8]. We then generated a panel of murine monoclonal antibodies specific to the naïve hPSCs by using primed hPSCs as a decoy immunogen as described previously [29]. We finally established a panel of hybridomas secreting MAbs which bound to the naïve H9 cells by flow cytometric analysis. The antibodies were further classified into naïve hPSC-specific, primed hPSC-specific, and both naïve- and primed-specific group according to their binding ability to primed hPSCs. Of the naïve hPSC-specific group, N1-A4 recognized an approximately 60 kDa cell surface-associated protein, which was identified as heat shock protein 60 (HSP60). HSP60 was expressed in primed hPSCs, but was rapidly downregulated to their differentiation derivatives during the embryoid body (EB) differentiation of hPSCs. HSP60 depletion led to a decrease in the expression of pluripotency genes in primed hPSCs. HSP60 depletion also decreased the expression of pluripotency genes and naïve-state-specific genes in naïve hPSCs, in which HSP60 was expressed on the surface. The results demonstrated that HSP60 is necessary for the maintenance of pluripotency genes in both primed and naïve hPSCs, and downregulated during the differentiation of hPSCs, suggesting that HSP60 is a regulator of hPSC pluripotency and differentiation. We also discuss how HSP60 is expressed on the cell surface and how it regulates stemness in hPSCs.

## Materials and methods

### Primed hPSC culture

H9 hPSC line was maintained on irradiated primary mouse embryonic fibroblasts (MEFs) in hPSC medium with DMEM/F12 (Welgene, Daegu, Korea) medium supplemented with 20% KnockOut Serum Replacement (KOSR, Thermo Fisher Scientific, Seoul, Korea), 0.1 mM MEM non-essential amino acids (MEM-NEAA, Welgene), 0.1 mM β-mercaptoethanol (Sigma-Aldrich, Seoul, Korea) and 4–8 ng/ml bFGF (R&D systems, Minneapolis, MN, USA) at 37˚C in 5% $O_2$ and 5% $CO_2$. CHA-hES4 was kindly provided by Dr. Hyung Min Chung (CHA University, Seoul, Korea) [30], and cultured as described above. mESC line R1 (ATCC, Manassas, VA, USA) was cultured as previously described [29]. To induce the ground state of mESCs, two inhibitors (2i, 1 μM PD0325901 and 3 μM CHIR99021) were added to the culture medium of R1 cells as described previously [31]. The human embryonal carcinoma cell lines, NT-2/D1 and 2102Ep, were cultured according to the instructions provided by ATCC. Cancer cell lines were obtained from ATCC and maintained according to the protocol provided by the supplier.

### Naive hPSC culture

Following routine culture with collagenase IV (1mg/ml), H9 hPSCs were plated as the small clumps on fresh inactivated MEFs in 2i/L/F/A naïve conversion medium [8]. 2i/L/F/A medium consisted of KO-DMEM (Thermo Fisher Scientific) supplemented with 20% KOSR, 0.1 mM β-mercaptoethanol, 12 ng/ml bFGF, 1000U recombinant human LIF (rhLIF, PrimeGene, Shanghai, China), 1 μM PD0325901 (Sigma-Aldrich), 3 μM CHIR99021 (Tocris, Seongnam, Korea), 10 μM Forskolin (Sigma-Aldrich), 50 ng/ml Ascorbic acid (Sigma-Aldrich) as described before [8]. After 4–6 days, the dome-shaped naive colonies were passaged as single cells using 0.05% trypsin/EDTA (Welgene) every 3 days and replated on freshly inactivated MEFs. 2i/L/F/A-driven naive hPSCs were cultured at 37˚C in 5% $O_2$ and 5% $CO_2$. For 2i/L/X/F/P-naive conversion, small clumps of H9 hPSCs were cultured in hPSC medium for 3 days and further cultured in 2i/L/X/F/P medium for 2 days as described before [17]. 2i/L/X/F/P medium consisted of DMEM/F12 (Thermo Fisher Scientific) supplemented with 20% KOSR, 0.1 mM β-mercaptoethanol, 10 ng/ml bFGF, 20 ng/ml rhLIF, 1 μM PD0325901, 3 μM CHIR99021, 4 μM XAV939 (Sigma-Aldrich), 10 μM Forskolin (Tocris) and 2 μM Purmorphamine (Tocris). After single cell passaging using accutase (Thermo Fisher Scientific), cells (50,000–100,000 cells/cm$^2$) were replated on freshly inactivated MEFs in 2i/L/X/F/P medium with 10 μM Y-27632. After 3–5 days, cells were passaged and replated cells with a density of 5,000–20,000 cells/cm$^2$ in 2i/L/X medium with 10 μM Y-27632 (Sigma-Aldrich), and without Forskolin and Purmorphamine for long-term maintenance. 2i/L/X/F/P cells were passaged as single cells using accutase every 3 days. 2i/L/X/F/P cells were cultured at 37˚C in 5% $O_2$ and 5% $CO_2$. For LCDM-naïve conversion, small clumps of H9 cells were cultured in LCDM medium [16]. LCDM medium consisted of 240 mL DMEM/F12, 240 mL Neurobasal (Thermo Fisher Scientific), 2.5 mL N2 supplement (Thermo Fisher Scientific), 5 mL B27 supplement (Thermo Fisher Scientific), 1% GlutaMAX (Thermo Fisher Scientific), 1% NEAA, 0.1 mM β-mercaptoethanol, penicillin-streptomycin, 5 mg/ml BSA (Sigma-Aldrich), and 5% KOSR, and it was supplemented with 10 ng/ml rhLIF (L), 1μM CHIR99021 (C), 2μM (S)-(+)-Dimethindenemaleate (D; Tocris), 2 μM Minocycline hydrochloride (M, Santa Cruz), 0.5–1 μM IWR-endo-1 (Selleckchem, Houston, TX, USA) and 2 μM Y-27632 (Sigma-Aldrich) [16]. LCDM medium was changed every day, and LCDM cells were passaged by single-cell trypsinization (0.05% trypsin-EDTA) every 3 days at a split ratio ranging from 1:3 to 1:5. LCDM cells were cultured under 20% $O_2$ and 5% $CO_2$ at 37˚C.

## Flow cytometry

Primed and naïve hPSC cultures were washed once in phosphate-buffered saline (PBS, pH7.4) and detached with collagenase (1mg/ml). MEFs were eliminated by pre-plating on gelatin-coated plates, and cell clumps were recovered and dissociated for 5 min at 37˚C with accutase. NT-2/D1, Huh7, SNU-387, and MEFs were harvested as single cell suspensions using trypsin/EDTA or enzyme-free dissociation solution. Single cell suspensions were filtered through a 40μm cell strainer. Cells were resuspended in PBA (1% bovine serum albumin, 0.02% NaN$_3$ in PBS, pH7.4). Cells were incubated for 20 minutes on ice with SSEA-3, SSEA-4, TRA-1-60, TRA-1-81, CD24, CD90, CD7, CD77, CD130, and N1-A4 antibodies. Isotype controls matching each immunoglobulin subtypes were also stained as control staining. After washing, propidium iodide (PI)-negative live cells were analyzed for antibody binding. Fluorescence detection was performed using a dual laser FACSCalibur flow cytometer (BD Biosciences, Seoul, Korea) and the BD CellQuest Pro analytical software (BD Biosciences).

## Embryoid body (EB) formation

H9 hPSCs were washed in hPSC medium without bFGF once and then incubate in collagenase IV (1 mg/ml) for 5 min at 37˚C to dissociate colonies to cell clumps. Cells were harvested and incubated on gelatin-coated plate in hPSC medium (without bFGF) for 15~30 min to remove MEFs. Cells were collected from the flasks and washed again with hPSC medium (without bFGF). hPSC clumps were seeded into bacterial dishes and cultured in DMEM/F12 medium supplemented with 20% fetal bovine serum (FBS), 0.1mM 2-mercaptoethanol, 1% non-essential amino acids, 100 U/ml penicillin G and 100 μg/ml streptomycin for 10 days. Medium was changed every other day.

## Antibody purification

MAbs were purified from the culture supernatant of hybridomas by Protein G-Agarose column chromatography as described previously [32, 33].

## Quantitative RT-PCR (qPCR)

Primed and naive hPSCs were pre-plated for 1 h on gelatin-coated plates to remove adherent MEFs. The total RNAs were isolated using the Trizol reagent (Thermo Fisher Scientific). RNAs were converted to cDNAs using PrimeScript™ RT reagent Kit (TaKaRa, Otsu, Japan). PCR was conducted using Power SYBR Green PCR Master Mix (Applied Biosystems, Waltham.MA, USA) on an ABI Prism 7000 Sequence Detection System. Relative gene expression levels between different samples were analyzed using the ΔΔCt method [34, 35]. The primers that were used for qPCR are listed in S1 Table.

## Cell surface biotinylation, immunoprecipitation, and Western blotting

Cell surface biotinylation of Huh7 cells was performed with EZ-Link Sulfo-NHS-LC-Biotin (Thermo Fischer Scientific) as described previously [34]. Briefly, biotin-labeled cells were lysed with 1% NP40 lysis buffer (25 mM Tris-HCl, pH 7.5, 250 mM NaCl, 5 mM EDTA, 0.1% Nonidet P-40, 2 μg/ml aprotinin, 100 μg/ml PMSF, and 2 μg/ml leupeptin) at 4˚C for 30 min. The cell lysates were clarified by centrifugation to remove nuclei at 4˚C, 12,000 rpm for 40 min before storage in −70˚C deep freezer. To remove the cellular proteins that nonspecifically bound to Protein G-Sepharose, the cell lysates were incubated with Protein G-Sepharose at 4˚C for 2 h, and the beads were then recovered and extensively washed with lysis buffer for use as a negative control (No Ab) for the immunoprecipitation experiments. For

immunoprecipitation of the antigens recognized by various antibodies, precleared lysates were incubated with approximately 2~4 μg N1-A4, goat-anti-HSP60 (K19, Santa Cruz Biotechnologies, Dallas, TX, USA), mouse anti-HSP60 (LK-1, Santa Cruz Biotechnologies) at 4°C overnight and further incubated with Protein G agarose. The beads were extensively washed with lysis buffer, and the bound proteins were eluted from the beads by boiling at 100°C for 5 min. Proteins were then transferred to a nitrocellulose membrane, and the membrane was blocked in 5% skim milk in PBST (PBS containing 0.1% Tween 20) at room temperature (RT) for 1 h. The membrane was then incubated with streptavidin-horse radish peroxidase (SA-HRP, GE Healthcare, Seoul, Korea) at RT for 1 h. The membrane was also incubated with goat anti-HSP60, mouse anti-HSP60, anti-His (Thermo Fisher Scientific) and/or anti-myc antibodies (Thermo Fisher Scientific). After extensive washing, the biotinylated proteins were visualized using enhanced chemiluminescence (ECL) detection reagent (GE Healthcare).

## Mass spectrometry

The protein of interest was enzymatically digested in-gel in a manner similar to that described previously [33]. The search program ProFound was used for protein identification [36].

## DNA and siRNA transfection

HSP60 cDNA was purchased from the Korea Human Gene Bank (KRIBB, Daejon, Korea) and cloned into the HindIII/XbaI sites of pcDNA3.1(+)-Myc-His expression vector (Thermo Fisher Scientific). 293FT ($3~4\times10^5$ cells/well) cells were transfected with pcDNA3.1(+) or pcDNA3.1(+)-HSP60-myc-His vectors, using Lipofectamine™ 2000 (Thermo Fischer Scientific) according to the supplier's protocol. Protein was transiently expressed for 48 h, and the cell pellet was lysed in 1% NP40 lysis buffer. Cell lysates were subjected to Western blot analysis or immunoprecipitation with anti-Myc (Thermo Fisher Scientific), anti-His (Thermo Fisher Scientific), or anti-HSP60 antibodies (Santa Cruz Biotechnologies). To knock down HSP60, H9 cells preincubated with 10 μm Y-27632 for 1 h were dissociated with TrypLE Select (Thermo Fisher Scientific), and $2\times10^5$ cells were seeded into Matrigel (BD Biosciences) coated-feeder free plate in TeSR-E8 medium (STEMCELL Technologies, Vancouver, Canada). The next day, the cells were transfected with HSP60 siRNA #1 and #2 (Bioneer, Daejon, Korea) or accutarget negative control siRNA (Bioneer) using RNAimax (Thermo Fisher Scientific) according to the supplier's protocol. The sense sequences of HSP60 siRNA #1, 2 were 5′-GUGUUGAAGGAUCUUU-GAU-3′ and 5′-GAAGUUUGAUCGAGGCUAU-3′, respectively. After transfection, the cells were incubated for 48 h before further analysis.

## Statistical analysis

The unpaired samples t-test was a statistical procedure used to compare the difference between two populations, and a p-value of less than 0.05 was considered statistically significant. Data were expressed as the mean ± standard deviation (SD).

## Results

### Induction of naive pluripotency from primed hPSCs by three different methods

H9 and CHA-hES4 hPSC cells were cultured on MEFs in three different media formulations, 2i/L/F/A [8], 2i/L/X/F/P [17] and LCDM [16]. Large and flat hPSC clumps were converted to small and dome-shaped clumps in all the three conversion media indicative of naive pluripotency (**Fig 1A**). The domed colonies were passaged with trypsin/EDTA or accutase, similar to

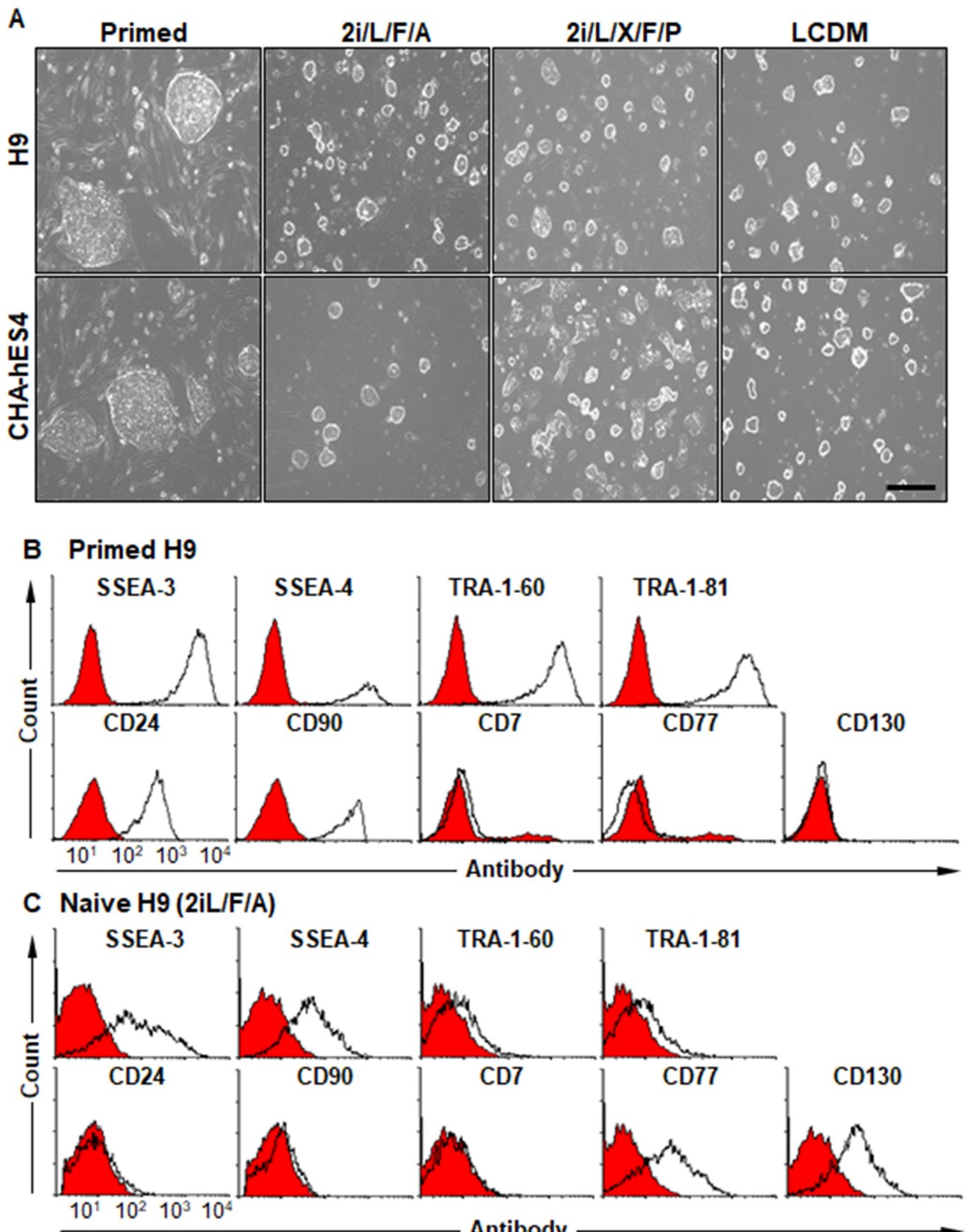

**Fig 1. Conversion of primed hESCs to the naïve state of pluripotency.** (A) Colony morphology of primed H9 and CHA-hES4 cells and their naïve cells cultured in 2i/LF/A, 2i/L/X/F/P or LCDM media. The scale bar is 100 μm. (B, C) Cell surface expression of primed (SSE-3, SSEA-4, TRA-1-60, TRA-1-81, CD24, and CD90) and naïve markers (CD7, CD77, and CD130) on primed H9 (B) and 2i/L/F/A-naïve hPSCs. (C). Black lines, cell surface marker with the indicated primary antibodies; red area, no primary antibody control.

how mESCs were split. To compare primed hPSCs and naïve hPSCs, the expression of SSEA-3, SSEA-4, TRA-1-60, TRA-1-81, CD24, and CD90, known as primed hPSC markers, was analyzed in primed and naïve H9 hPSCs cultured in 2i/L/F/A medium by flow cytometry. The expression of CD7, CD77, and CD130, known as naïve hPSC markers, was also observed in both hPSCs. The expression of primed hPSC markers, TRA-1-60, TRA-1-81, CD24, and CD90, rapidly decreased in naïve hPSCs (**Fig 1B and 1C**). The relative expression of SSEA-3 and SSEA-4 was also significantly decreased in naïve hPSCs, confirming that naïve hPSCs were induced by the 2i/L/F/A medium. Furthermore, the 2i/L/F/A medium induced the expression of CD77 and CD130, naïve hPSC markers, although it did not induce the expression of CD7 (**Fig 1B and 1C**).

In addition, qPCR also showed that the expression of naïve-state-specific genes (*STELLA*, *ESRRB*, *PRDM14*, *REX1*, *KLR2*, *KLF4*, *KLF5*, *DNMT3L*, *TBX3*, *DPPA5*, *XIST*) was increased in naïve hPSCs, while the expression of the primed state-specific genes (*OTX2*, *DNMT3B*) was decreased (**S1 Fig** and **S1 Table**). Therefore, the results confirm that the 2i/L/F/A medium properly induced naïve hPSCs from primed hPSCs.

## Generation of a panel of monoclonal antibodies against naïve hPSCs

To generate a panel of MAbs that bound to naïve hPSCs cultured in the 2i/L/F/A medium but not to primed hPSCs, primed H9 cells were first immunized into the right hind footpads of BALB/c mice as decoy immunogen, and then the naïve H9 cells were immunized into the left hind footpads 3 days later as described previously [29]. The lymphocytes from the left hind popliteal lymph nodes were fused to FO myeloma cells, and the fusion generated a panel of hybridomas secreting MAbs that bound to the naïve hPSCs. The antibodies were further classified into naïve hPSC-specific, primed hPSC-specific, and both naïve- and primed-specific group according to their binding ability to primed hPSCs. Of the naïve hPSC-specific group, N1-A4 bound to the naïve H9 hPSCs cultured in 2i/L/F/A and LCDM media but not to the naïve hPSCs cultured in 2i/L/X/F/P medium (**Fig 2A**). For CHA-hES4 hPSCs, another conventional primed hPSCs, N1-A4 bound to naïve hPSCs cultured in 2i/L/F/A and 2i/L/X/F/P but not to naïve hPSCs cultured in LCDM (**Fig 2A**). Thus, N1-A4 antigen was expressed on the surface of two different naïve hPSC lines cultured in 2i/L/F/A medium. N1-A4 did not bind to the human embryonal carcinoma lines (NT-2/D1 and 2102Ep) and MEFs, but it bound to the hepatocellular carcinoma (HCC) cell lines, Huh7 and SNU-387 (**Fig 2B**). Taken together, the results suggest that N1-A4 antigen is expressed on the surface of some naïve hPSCs and HCC cell lines.

## N1-A4 recognizes cell surface-expressed HSP60 in a conformational dependent manner

To identify the cell surface antigen recognized by N1-A4, surface proteins of Huh7 cells were biotinylated, the cell lysates were immunoprecipitated with N1-A4, and the immunoprecipitates were then visualized with SA-HRP. N1-A4 immunoprecipitated an approximately 60 kDa surface protein on Huh7 cells (**S2A Fig**). The same proteins were excised and subjected to mass spectrometry, and were identified as HSP60 from a protein database search (**S2B and S2C Fig**). To confirm whether N1-A4 recognizes HSP60, Huh7 cell lysates were

immunoprecipitated with N1-A4 and a commercially available goat anti-HSP60 antibody. The immunoprecipitates were detected at approximately 60 kDa by Western blotting with the goat anti-HSP60 antibody, suggesting that N1-A4 immunoprecipitates HSP60 (Fig 3A). However, the HSP60 protein was not detected by Western blotting with N1-A4 (S3A Fig). When Huh7 cell lysates were immunoprecipitated with N1-A4 and a commercially available mouse mono-clonal anti-HSP60 antibody, the 60kDa HSP60 protein was immunoprecipitated with N1-A4 but not with the mouse anti-HSP60 antibody (S3B Fig). The results indicate that N1-A4 recognizes cell surface-expressed HSP60 in a conformation-dependent manner, contrary to the commercial anti-HSP60 antibodies. To further demonstrate that N1-A4 recognizes HSP60, control and Myc/His-tagged HSP60 expression plasmids were transfected into HEK293FT cells, and the cell lysates were subjected to immunoprecipitation with anti-Myc, anti-HSP60, and N1-A4. Myc-tagged HSP60 proteins were detected in HSP60-transfected lysates with all three antibodies (Fig 3B and 3C). When the cell lysates were subjected to immunoprecipita-tion with anti-Myc, anti-His and N1-A4, Myc-tagged HSP60 proteins were detected in

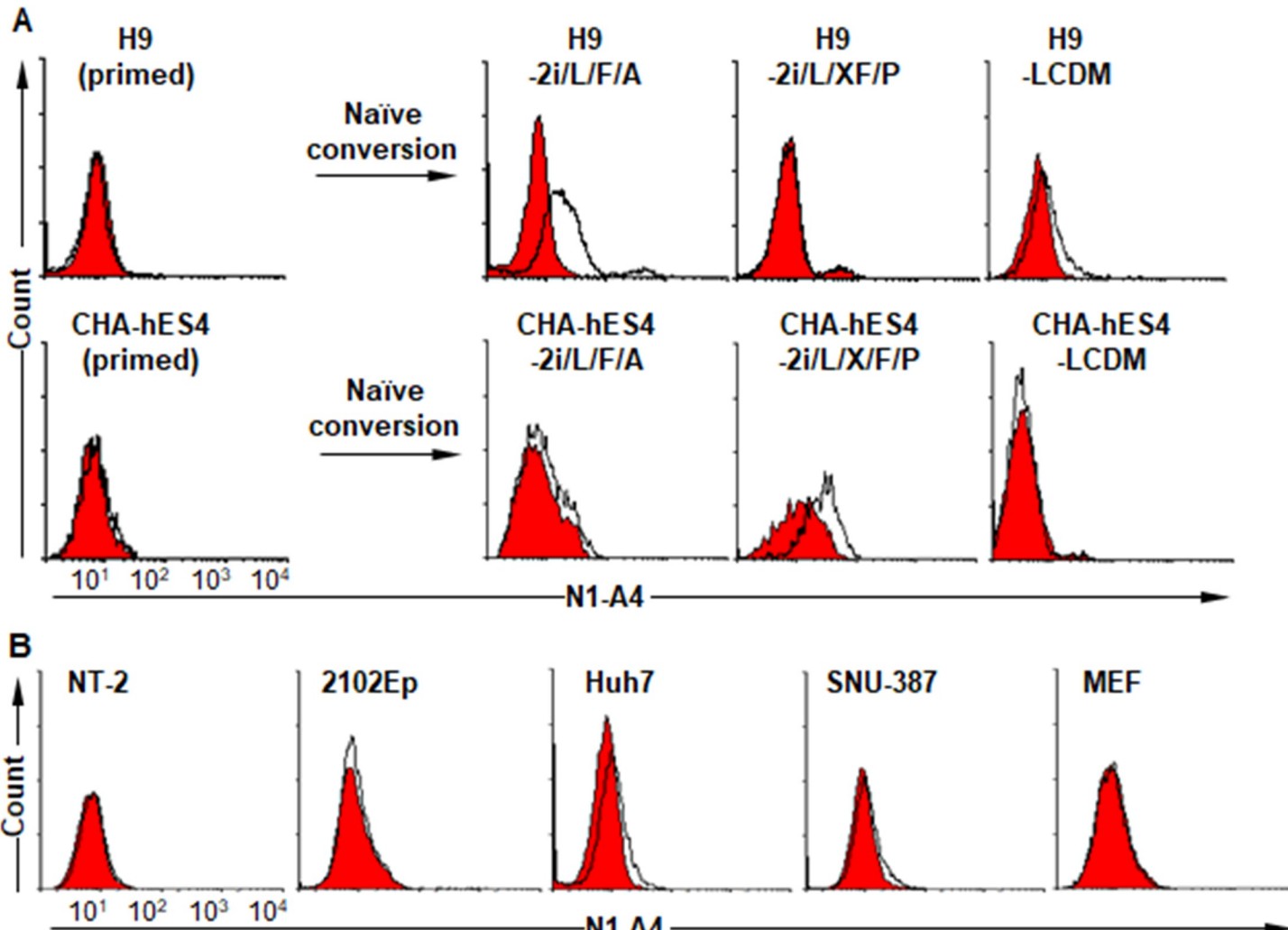

**Fig 2. Binding profiles of N1-A4 on primed/naïve hPSCs and various cells.** (A, B) Expression of N1-A4 antigen was examined by flow cytometry in primed and naïve hPSCs (A), human embryonic carcinoma cells (NT-2/D1, 2102Ep), hepatocellular carcinoma cells, (Huh7, SNU-387), and MEFs (B). Black lines, cell surface marker with the indicated primary antibody; red area, no primary antibody control.

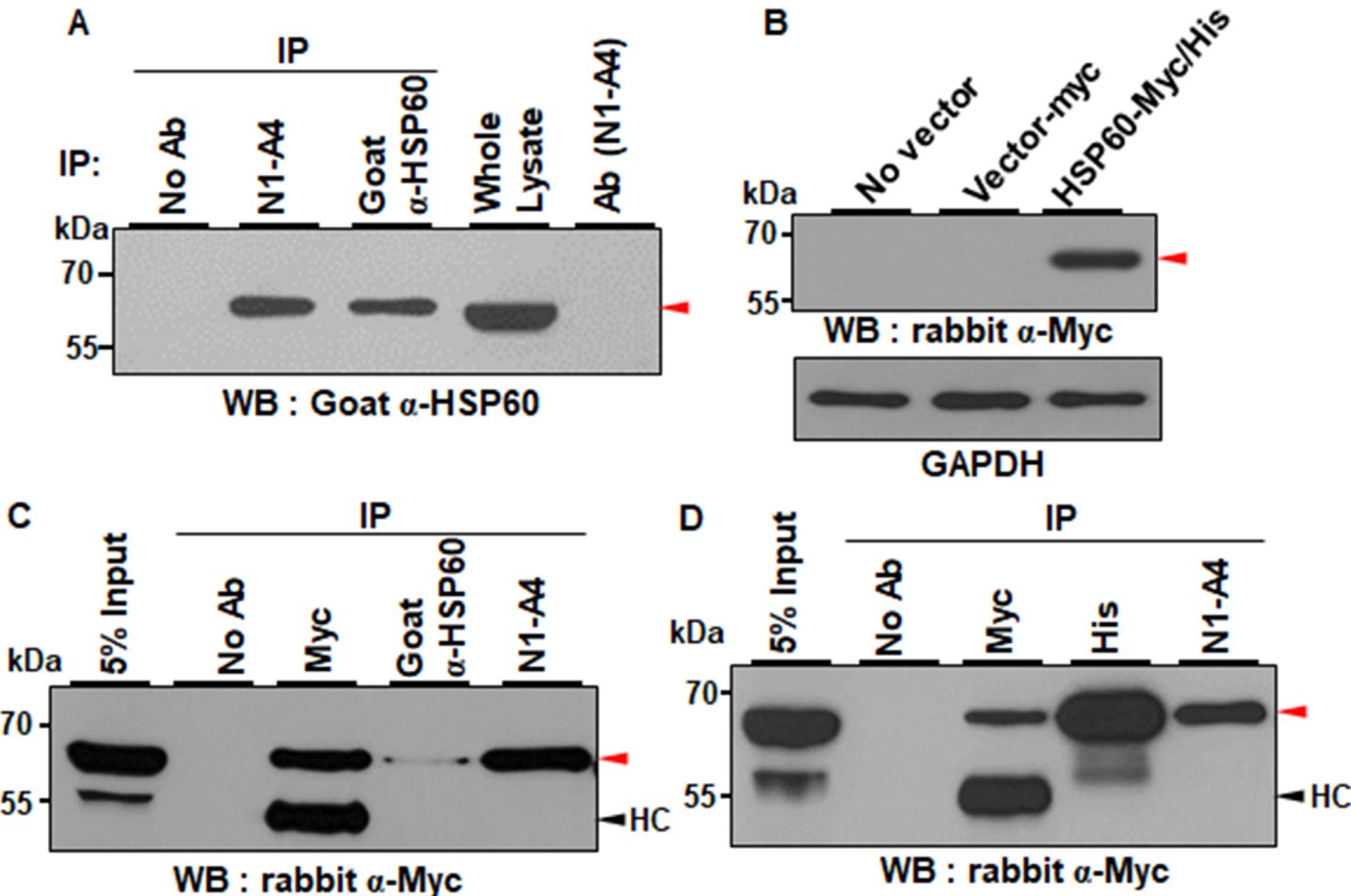

**Fig 3. N1-A4 recognizes HSP60.** (A) Huh7 cell lysates were immunoprecipitated with N1-A4 or goat anti-HSP60 antibodies (α-HSP60), and the immunoprecipitates were detected with α-HSP60 in Western blot analysis. Red arrowhead indicates HSP60. (B) HSP60-Myc/His vector was overexpressed in 293FT cells, and Myc-tagged HSP60 protein was detected with α-Myc. β-actin expression was the loading control. (C, D) HEK293FT cells were transfected with HSP60-Myc/His expression vector. Cell lysates were immunoprecipitated with α-Myc, α-HSP60, and N1-A4 (C), or α-Myc, α-His, and N1-A4 (D). The immunoprecipitates were detected by Western blot analysis with α-Myc. Red arrowhead indicates the HSP60 proteins.

HSP60-transfected lysates with all three antibodies (Fig 3D), indicating that N1-A4 antigen is HSP60 indeed. Taken together, the result demonstrates that N1-A4 recognizes cell surface expressed HSP60 in a conformation-dependent manner.

## HSP60 is necessary for the maintenance of primed pluripotency in hPSC

Many chaperones are associated with the maintenance of pluripotency in pluripotent stem cells through the control of protein homeostasis [37]. HSP60 depletion decreases OCT4 expression in mESCs, increasing apoptosis during mESC differentiation into EBs [38], suggesting that HSP60 is required for PSC stemness and differentiation. To examine whether HSP60 expression is altered during differentiation of primed hPSCs, HSP60 expression was estimated during EB formation of primed hPSCs by qPCR. As expected, *OCT4*, a pluripotency gene, was downregulated during the EB differentiation of primed hPSCs, and *PAX6*, an early differentiation gene, was upregulated (Fig 4A and 4B). As with *OCT4*, *HSP60* was also down-regulated during the EB differentiation of hPSCs, although it was slightly recovered after 7 days of differentiation (Fig 4C).

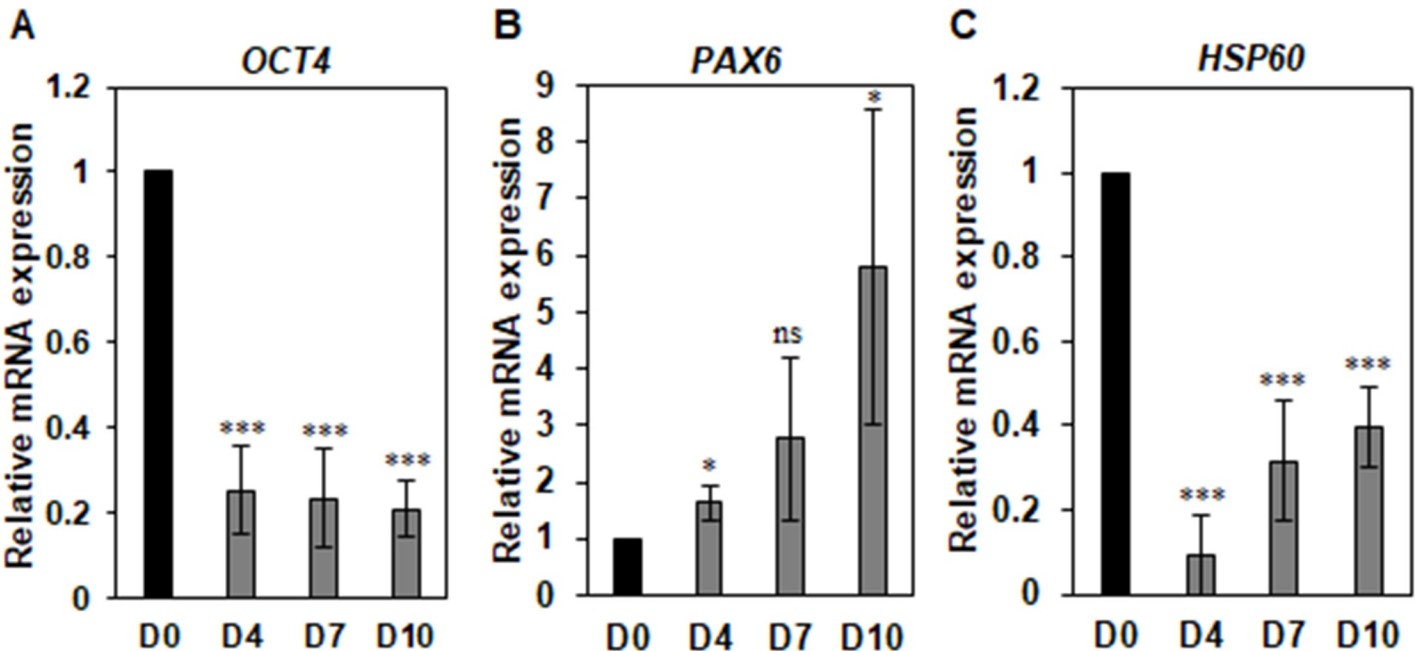

**Fig 4. Expression kinetics of HSP60 during EB differentiation of primed hPSCs.** Primed H9 hPSCs were differentiated by EB formation for 10 days. The expression of *OCT4*, *PAX6*, and *HSP60* genes was determined by qPCR. qPCR data are presented as relative expression changes considering expression in undifferentiated hPSCs (Day 0) as 1. The graphs represent the mean values of three independent determinations ±SD (ns, not significant; *, p < .05; ***, p < .005).

The result suggests that HSP60 is expressed in hPSCs but downregulated in their differentiated progenies. Therefore, we next examined whether HSP60 is required for the maintenance of pluripotency in primed hPSCs. To study the role of HSP60 in primed hPSCs, we depleted HSP60 in primed H9 hPSCs by two HSP60 siRNAs (**S4 Fig**). HSP60 protein expression was downregulated up to approximately 46–73% by siRNA2 transfection (**S4 and 5 Figs**). Western blot analysis showed that HSP60 depletion led to a decrease in the expression of OCT4, NANOG, and SOX2 by approximately 48%, 12%, and 68%, respectively (**Fig 5A**). qPCR analysis further showed that HSP60 depletion led to a decrease in the expression of OCT4, NANOG and SOX2 mRNAs as well (**Fig 5C–5E**). Thus, HSP60 expression is required for the maintenance of primed pluripotency in hPSC.

## HSP60 depletion impairs naïve pluripotency in hPSCs

Since we observed HSP60 expression on the surface of naive hPSCs (**Fig 2A**), we investigated the cell surface expression of HSP60 in mESCs, a mouse version of naive hPSCs. To induce the ground state of mESCs, 2i (mitogen activated protein kinase and glycogen synthase kinase 3β inhibitor)-treated mESCs were also analyzed [31]. Since the amino acid sequence homology between the two species of mouse and human reaches 98% in the multiple sequence alignment analysis [39], we predicted that N1-A4 is able to recognize cell surface-expressed HSP60 on mESCs as well. However, HSP60 was not detected on mESCs and 2i-treated mESCs (**S5 Fig**), suggesting that cell surface translocation of HSP60 may be a unique phenomenon seen in some human naive hPSCs (**Fig 2**). To study the role of HSP60 in naïve pluripotency, we depleted HSP60 in naïve hPSCs cultured in the 2iL/F/A medium by the same siRNA2 used in primed hPSCs. HSP60 expression was downregulated up to approximately 67% by siRNA2 transfection (**Fig 6A**). qPCR analysis showed that HSP60 depletion led to a decrease in the expression of *OCT4* and *NANOG* (**Fig 6B and 6C**). To further investigate the role of HSP60 in naïve

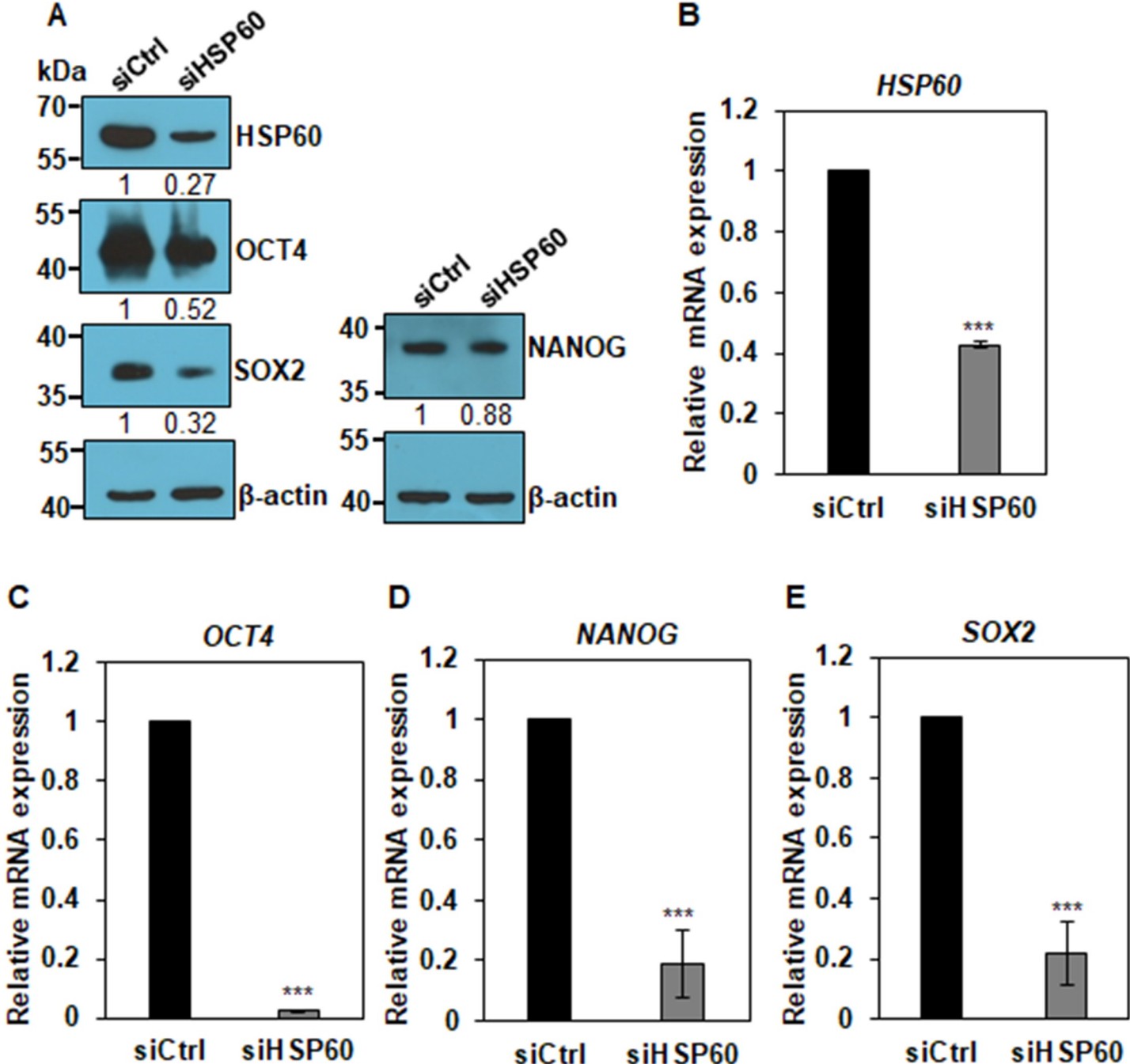

**Fig 5. Knockdown effects of HSP60 in primed hPSCs.** (A) Cell lysates from HSP60 siRNA-transfected H9 hPSCs were subjected to Western blot analysis with antibodies against HSP60, OCT4, SOX2, and NANOG. Relative protein levels were measured using ImageJ software and normalized to the β-actin. (B) qPCR analysis for the expression of HSP60 and pluripotency genes (*OCT4*, *SOX2*, and *NANOG*) in control (siCtrl) or HSP60 siRNA-transfected H9 hPSCs (siHSP60). Relative mRNA levels were measured by qPCR and were shown after normalization against *GAPDH* mRNAs. The graphs represent the mean values of at least two independent determinations ±SD (***, p < .005).

pluripotency, the expression levels of naïve-state-specific genes were also examined in naïve hPSCs by qPCR analysis. HSP60 depletion led to a decrease in the expression of naïve-state-specific genes, *PRDM14*, *KLF4*, and *DPPA5* (**Fig 6E, 6F and 6H**) [8, 9, 14, 40]. HSP60 depletion

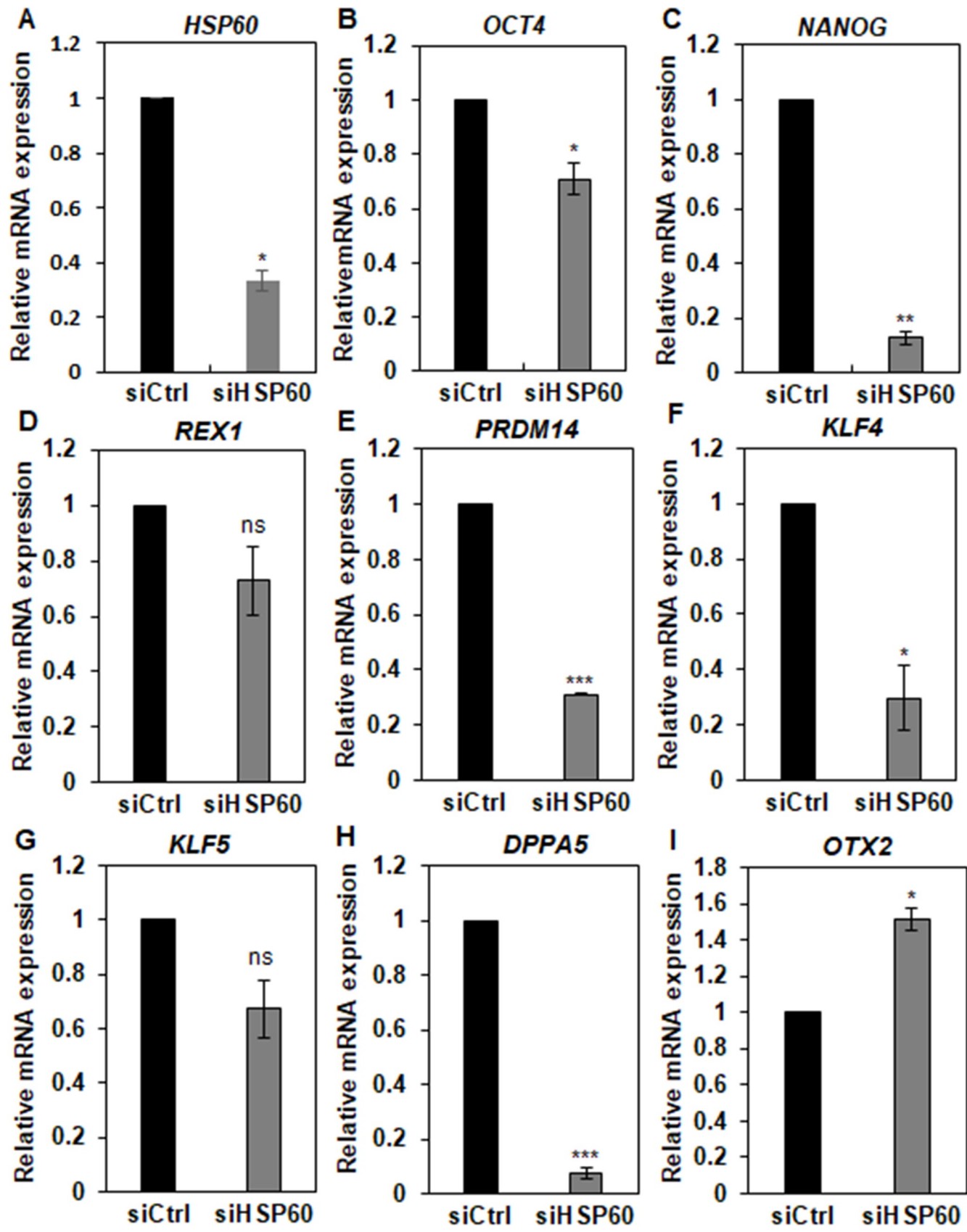

**Fig 6. Knockdown effects of HSP60 in naïve hPSCs.** The relative expression of pluripotency genes (*OCT4 and NANOG*), naïve-state-specific genes (*REX1*, *PRDM14*, *KLF4*, *KLF5*, and *DPPA5*), primed-state-specific genes (*OTX2*), and *HSP60* genes was measured in siHSP60-transfected 2i/L/F/A-naïve hPSCs by qPCR. qPCR data are presented as relative expression changes, considering expression in siCtrl-transfected hPSCs as 1. The graphs represent the mean values of two independent determinations ±SD (ns, not significant; *, p < .05; **, p < .01; ***, p < .005).

also led to an increase in the expression of *OTX2*, a primed-state-specific gene (**Fig 6I**) [9, 40], suggesting that HSP60 is required for the maintenance of naïve pluripotency in naïve hPSCs.

## Discussion

HSP60 is mainly located in the mitochondria but is also found in the cytosol, on the cell surface, and in the extracellular space [41, 42]. The present study showed that it is not expressed on the surface of primed hPSCs, but it is expressed on the surface of naïve hPSCs cultured in 2i/L/F/A, 2i/L/X/F/P, and LCDM media (**Fig 2A**) [8, 16, 17]. Thus, HSP60 is expressed on the surface of some naïve hPSCs. Some HSPs containing HSP70 and HSC70 are also detected on the surface of primed hPSCs but not on the differentiated derivatives [43, 44]. However, there is no report on HSP60 on naïve hPSCs. Thus, the present study is the first report on cell surface expression of HSP60 on naïve hPSCs. Biological significance of surface expression of HSP60 in naïve hPSCs is not clear. HSP60 is identified in the natural environment of undifferentiated hPSCs, namely, the blastocoel fluid, which is in contact with all the cells in the blastocyst [45]. HSPs containing HSP60 are essential for successful preimplantation development, and the specific expression pattern of HSPs may play both an essential role for differentiation and a protective role against apoptosis in the developing embryo [46]. Furthermore, anti-HSP70 treatment increases apoptosis and reduces the number of embryos reaching the blastocyte stage during the development of bovine embryos in vitro [47]. HSP60 depletion also enhances apoptosis during mESC differentiation into EBs, indicating that HSP60 promotes survival of differentiated cells from mESCs [38]. Many HSPs are also regarded as regulators of apoptosis [48]. In this regard, cell surface-expressed HSP60 may take part in the regulation of apoptosis in naïve hPSCs. Further studies are needed to understand how and why HSP60 is expressed on the surface of naïve hPSCs.

Cultured PSCs require continuous protein synthesis for cell maintenance and proliferation. To maintain protein homeostasis and suppress protein abnormalities and aberrations in PSC biology, enhanced mechanisms of proteome quality control is needed through elevated expression of molecular chaperones [37]. Many molecular chaperones are closely associated with pluripotency maintenance and proper differentiation in PSCs. HSP60 is a molecular chaperone belonging to the Chaperonins of Group I and functions mainly inside mitochondria, in which it maintains protein homeostasis [49]. HSP60 also plays in many other cellular compartments, such as cytosol, plasma membrane, extracellular space, and body fluids. It is typically associated with the synthesis and transportation of mitochondrial proteins from the cytoplasm to the mitochondria [49]. HSP60 has been known to promote self-renewal, proliferation and survival in mESCs through OCT4 expression [38]. As shown in mESCs, HSP60 protein level is also high in rabbit ESCs when compared to differentiated cells, suggesting that HSP60 might have an important role in maintaining the undifferentiated status of general ESCs [50]. The present study found that HSP60 is required for the maintenance of pluripotency in primed hPSCs and is also positively associated with the maintenance of naïve pluripotency in naïve hPSCs through the control of some naïve-state-specific genes (**Figs 4–6**). The data suggest that HSP60 plays a similar role in both primed and naïve hPSCs. Two pluripotency states are regarded as quite different statuses of pluripotency [3, 4], but the other study also suggests that the heterogeneity between two states is expressed in low levels [51]. The main role of HSP60 is to regulate the folding and trafficking of other proteins in mitochondria and cytosol. HSP60

has been suggested to associate with some pluripotency proteins (OCT4, NANOG, c-myc and Stat3) [50]. The present study also found that HSP60 depletion decreases the expression of *OCT4* and *NANOG* in both primed and naïve hPSCs. Therefore, it is possible to speculate that HSP60 may bind and regulate common proteins in both primed and naïve hPSCs.

HSP60 expression is high in mESCs, rabbit ESCs, and mouse embryonal carcinoma stem cells as compared with their differentiated derivatives, but decreases with differentiation [38, 50, 52, 53]. The present study also shows that HSP60 expression is high in conventional primed hPSCs as compared with their differentiated derivatives but decreases with differentiation. Thus, HSP60 appears to be a negative differentiation marker in PSCs. When inducing differentiation in mESCs, HSP60 knockdown does not change the expression of nestin, an ectoderm marker, but increases the expression of α-fetoprotein, an endoderm marker, and decreases the expression of brachyury, a mesoderm marker [38]. HSP60 protein expression is also decreased following EB formation and neural differentiation in P19 mouse embryonal carcinoma stem cells [53]. Thus, HSP60 participates in the regulation of differentiation of PSCs. However, the exact role of HSP60 in PSC differentiation is not clearly understood and further studies are needed.

N1-A4 was not able to detect cell surface-expressed HSP60 on mESCs, a mouse version of naïve state (**S5 Fig**). The result was not expected because the amino acid sequence homology between mouse and human reaches 98% in the multiple sequence alignment analysis. One possible explanation is that this phenomenon may be due to different molecular conformation between human and mouse. Cell surface translocation of HSP60 may undergo additional post-translational modifications or distinct conformational changes because the cell surface-expressed HSP60 is phosphorylated, N-glycosylated or citrullinated [49]. Immunoprecipitation and Western blot analysis using commercial anti-HSP60 antibodies and N1-A4 confirmed that N1-A4 recognizes cell surface-expressed HSP60 in a conformation-dependent manner, contrary to commercial anti-HSP60 antibodies (**Figs 3A and S3**). Therefore, it is likely that N1-A4 recognizes a natural physiological epitope of cell surface-expressed HSP60 on naïve hPSCs, which is different from that of mESCs. Further studies are needed to understand the differences between mouse and human naïve hPSCs.

In conclusion, a MAb, N1-A4, recognizes HSP60 on the surface of naïve hPSCs in a conformation-dependent manner. Subsequent analysis revealed that HSP60 is downregulated during differentiation of hPSCs, and is required for the maintenance of pluripotency genes in both primed and naïve hPSCs. The results suggest that HSP60 is a common regulator of hPSC pluripotency in both primed and naïve hPSCs.

## Supporting information

**S1 Table. Primers used for the qPCR analysis.**
(DOCX)

**S1 Fig. qPCR analysis for representative primed- and naïve-state-specific genes in primed and naïve hPSCs.** Representative primed- (*OTX2* and *DNMT3B*) and naïve-state-specific genes (*STELLA*, *ESRRB*, *PRDM14*, *REX1*, *KLF2/4/5*, *DNMT3L*, *TBX3*, *DPPA5* and *XIST*) were selected, and their expression was analyzed by qPCR. Shown is the relative expression of selected genes in 2i/L/F/A-naïve hPSCs compared to primed hPSCs. The graph represents the mean values of two independent determinations ±SD.
(TIF)

**S2 Fig. Antigen identification of N1-A4.** (A) Cell surface proteins of Huh7 cells were biotinylated, and the cell lysates were subjected to immunoprecipitation with N1-A4. Immunoprecipitates

were detected with streptavidin-horse radish peroxidase (SA-HRP). (B) Immunoprecipitates were stained with PageBlue. The 60 kDa protein immunoprecipitated by N1-A4 is indicated by red square. (C) Mass spectrometry analysis of the 60 kDa immunoprecipitate. The 60 kDa protein band was excised and analyzed by LC-MS/MS. Eighteen tryptic peptides (bold red) originating from the 60 kDa protein matched HSP60.
(TIF)

**S3 Fig. N1-A4 recognizes HSP60 in a conformational dependent manner.** (A) Huh7 cell lysates were immunoprecipitated with N1-A4 or goat α-HSP60, and the immunoprecipitates were detected with N1-A4 in Western blot analysis. LC, immunoglobulin light chain. (A) Huh7 cell lysates were immunoprecipitated with N1-A4 or mouse anti-HSP60 antibodies (α-HSP60), and the immunoprecipitates were detected with mouse α-HSP60 in Western blot analysis. Red arrowhead indicates HSP60. HC, immunoglobulin heavy chain; LC, immuno-globulin light chain.
(TIF)

**S4 Fig. Knockdown efficiency of two HSP60 siRNAs in primed H9 hPSCs.** (A) Two HSP60 siRNAs were tested for HSP60 knockdown in primed H9 hPSCs. Relative mRNA levels of HSP60 were measured by qPCR and were shown after normalization against GAPDH mRNA. The graphs represent the mean values of four independent determinations ± SD (***, p < .005). (B) HSP60 proteins were analyzed in HSP60 knockdown hPSCs by Western blot analysis. Relative protein levels of HSP60 were measured using ImageJ software and normalized to the β-actin.
(TIF)

**S5 Fig. Binding activity of N1-A4 in 2i-treated ground state mESCs.** (A) Cell morphology of mESC R1 and 2i-treated R1 cells. (B) Flow cytometry analysis of SSEA-1 and N1-A4 in mESC R1 and 2i-treated R1 cells. Purple and black lines are the binding activity of the indicated primary antibody in 2i-treated or mESC R1 cells respectively. Red area represents no primary antibody control.
(TIF)

**S1 Raw images.**
(PDF)

## Acknowledgments

We thank Dr. Hee Chul Lee for his comments and proofreading the manuscript.

## Author Contributions

**Conceptualization:** Hong Seo Choi, Chun Jeih Ryu.

**Data curation:** Hong Seo Choi, Hyun Min Lee, Min Kyu Kim, Chun Jeih Ryu.

**Formal analysis:** Hong Seo Choi, Hyun Min Lee, Min Kyu Kim, Chun Jeih Ryu.

**Funding acquisition:** Chun Jeih Ryu.

**Investigation:** Hong Seo Choi, Chun Jeih Ryu.

**Methodology:** Hong Seo Choi, Hyun Min Lee, Min Kyu Kim, Chun Jeih Ryu.

**Supervision:** Chun Jeih Ryu.

**Validation:** Chun Jeih Ryu.

**Writing – original draft:** Chun Jeih Ryu.

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
