## [Decision Letter · Decision Letter 0]

9 May 2022

PONE-D-22-07296Dept. of Integrative Bioscience and Biotechnology, Sejong UniversityPLOS ONE

Dear Dr. Ryu,

Thank you for submitting your manuscript to PLOS ONE. After careful consideration, we feel that it has merit but does not fully meet PLOS ONE’s publication criteria as it currently stands. Therefore, we invite you to submit a revised version of the manuscript that addresses the points raised during the review process.Specifically, please provide your responses to the reviewer's comments on your discussion section.

We look forward to receiving your revised manuscript.

Kind regards,

Johnson Rajasingh, Ph.D, HCLD

Academic Editor

PLOS ONE

Journal Requirements:

Additional Editor Comments (if provided):

Reviewers' comments:

Reviewer's Responses to Questions

**Comments to the Author**

1. Is the manuscript technically sound, and do the data support the conclusions?

Reviewer #1: Yes

2. Has the statistical analysis been performed appropriately and rigorously? 

Reviewer #1: Yes

3. Have the authors made all data underlying the findings in their manuscript fully available?

Reviewer #1: Yes

4. Is the manuscript presented in an intelligible fashion and written in standard English?

Reviewer #1: Yes

5. Review Comments to the Author

Reviewer #1: The manuscript number PONE-D-22-07296

The authors carryout excellent work about importance of HSP60 in pluripotency primed and naive states. This types of research work is essential for regenerative medicine. The authors need to add some more points in discussion part, about while on differentiation states what is status of HSP60 and how it is regulates the differentiation of various cells. The manuscript scientifically very sound and well written.

6. PLOS authors have the option to publish the peer review history of their article (what does this mean?). If published, this will include your full peer review and any attached files.

Reviewer #1: **Yes: **Dr. Kanagaraj Palaniyandi

---

## [Author Response · Author response to Decision Letter 0]

17 May 2022

Point-to-point replies

Journal Requirements:

Response) We checked whether all references are appropriate or not

Reviewers' comments:

Reviewer #1: The manuscript number PONE-D-22-07296

The authors carryout excellent work about importance of HSP60 in pluripotency primed and naive states. This types of research work is essential for regenerative medicine. The authors need to add some more points in discussion part, about while on differentiation states what is status of HSP60 and how it is regulates the differentiation of various cells. The manuscript scientifically very sound and well written.

Response) We added some more points about the role of HSP60 in PSC differentiation in discussion part. The changes that we made are red-colored in the revised manuscript with track changes.

---

## [Editor Report · Decision Letter 1]

24 May 2022

Role of heat shock protein 60 in primed and naïve states of human pluripotent stem cells

PONE-D-22-07296R1

Dear Dr. Ryu,

We’re pleased to inform you that your manuscript has been judged scientifically suitable for publication and will be formally accepted for publication once it meets all outstanding technical requirements.

Kind regards,

Johnson Rajasingh, Ph.D, HCLD

Academic Editor

PLOS ONE
---

## [Editor Report · Acceptance letter]

30 May 2022

PONE-D-22-07296R1 

Role of heat shock protein 60 in primed and naïve states of human pluripotent stem cells 

Dear Dr. Ryu:

I'm pleased to inform you that your manuscript has been deemed suitable for publication in PLOS ONE. Congratulations! Your manuscript is now with our production department. 

Kind regards, 

on behalf of

Dr. Johnson Rajasingh 

Academic Editor

PLOS ONE